# Economic, Environmental and Social Benefits of Adoption of Pyrolysis Process of Tires: A Feasible and Ecofriendly Mode to Reduce the Impacts of Scrap Tires in Brazil

**Geraldo Cardoso de Oliveira Neto [1,2,\*]**, **Luiz Eduardo Carvalho Chaves [3]**,
**Luiz Fernando Rodrigues Pinto [1]**, **José Carlos Curvelo Santana [1]**,
**Marlene Paula Castro Amorim [2,\*]** and **Mário Jorge Ferreira Rodrigues [4]**

1   Industrial Engineering Post-Graduation Program, UniversidadeNove de Julho (UNINOVE), Vergueiro Street, 235/249 – 12º, Liberdade, São Paulo 01504-001, Brazil; lfernandorp44@gmail.com (L.F.R.P.); jccurvelo@uni9.pro.br or jccurvelo@yahoo.com.br (J.C.C.S.)
2   GOVCOPP & Departamento de Economia Gestão e Engenharia Industrial e Turismo, Universidade de Aveiro, 3810-193 Aveiro, Portugal
3   School of Technology of São Paulo, FATEC, Tiradentes Avenue, São Paulo 01124-060, SP, Brazil; luizchaves@hotmail.com
4   IEETA & ESTGA, Universidade de Aveiro, 3810-193 Aveiro, Portugal; mjfr@ua.pt
\*   Correspondence: geraldo.prod@gmail.com (G.C.d.O.N.); mamorim@ua.pt (M.P.C.A.)

**Abstract:** This study addressed the development of a pilot plant for pyrolysis of scrap tires to obtain carbon black and other byproducts. The work was motivated by the goal of contributing to the development and dissemination of knowledge about existing technologies that allow modern economies to transform waste into valuable products, by documenting and discussing an empirical application in Brazil. Thispaper describes the development of a market for steel scrap, pyrolytic oil and carbon black products obtained from a vacuum pyrolysis process. The research work was conducted in Brazil, and was guided by the twofold purpose of reducing the environmental impacts, while gaining economical sustainability. Modern economies increasingly need to devise strategies to address energy generation while preserving natural ecosystems. These strategies include leveraging the use of renewable energy sources. Acknowledging that scrap tires hold an enormous potential as a sustainable energy option, this study aimed to contribute to the development and maturity of eco-friendly processing approaches to realize its full potential. The work involved a preliminary phase concerned with the operation of vacuum pyrolysis of scrap tires at a laboratorial scale, followed by the design of the pilot plant that operated for 10 years, at the time of the study, with a 100 kg/h batch flow. Results show that the yield of the pyrolysis process was 41% pyrolytic oil, 38% carbon black, 12% gas, and 8.9% steel scrap, with a calorific value of 36 MJ/kg per tire. The carbon black was composed of 90% carbon, and the pyrolytic oil was composed of 66% gasoline and 33% other oils, which have higher quality and can be commercialized with a potential profit over 3 million dollars/year.

**Keywords:** carbon credit; eco-efficiency; environmental cost accounting; pyrolysis; solid waste

## 1. Introduction

### 1.1. The Problem of Scrap Tires

The management of waste associated with end-of-life products is a public concern and a priority of policy debates. The end-of-life scrap tires represent an important issue notably for the implications they hold for public health and environment. Tires are made of non-biodegradable material, i.e. styrene-butadiene rubber (SBR), and for this reason there is a growing concern among developed economies about the need to prevent harmful environmental effects associated with tire waste. One option for improving the management of the scrap tires is processing end-of-life items for recovering its material. The recovery of scrap tires is done by rubber crumb (fine-grained or granulated tire rubber) and removal of fiber and steel elements, and by using the resulting outputs for new rubber products, such as playgrounds, sports surfacing, and rubber-modified asphalt. An alternative for addressing end-of-life waste tires is subjecting them to processes that yield outputs that are suitable for energy applications [1].

The amount of scrap tires is increasing significantly as result of rapid economic growth and development of the transport industry. According to statistics reported by Wang et al. [2] and Li et al. [3], approximately 3.0 billion tires are generated globally, with a predicted growth rate of at least 1.0 billion tires each year. North America, Europe, and Asia generate many scrap tires, accounting for almost 90% of global tire production [4,5]. The estimated growth for the worldwide tire demand is about 4.3% per year, and reached 2.9 billion units in 2017, while waste tire disposal in 2015 reached nearly 1 billion units [6–8]. In this scenario, recycling end-of-life tires becomes an urgent need because the accumulation of discarded tires holds serious environmental risks. While some end-of-life tires are recapped or ground for particular reuses, a great volume is simply dumped in rural areas or in landfills. When buried in landfills, they can eventually float to the surface. In piles, the non-biodegradable rubber can cause serious harm if ignited. Likewise, tires infested with mosquitoes are a subject of increasing concern [2,5].

Most countries do not yet have specific legislation concerning the collection of waste tires, and those that do are still researching how to increase their life cycle, as succeeded in Poland with the implementation of the urban solid waste law in 2012 [9]. In Brazil, an urban solid waste law was implemented in 2010 [10] andTurkey implemented the urban solid waste law in 2014 [11], Even in countries such as Spain, which implemented such laws in 2005, problems persist concerning the correct final destiny of the waste tires [12]. In contrast, countries such as USA, Germany, and Japan have been deploying this technology since the 1970s [13], e.g. New York has usedthe unserviceable tires for the generation of energy and production of gas, fuel oil, and carbon black since 1990 [14].

The European Union (EU) has prohibited the disposal of tires in landfills since 2009, and the recycling rate is 95% for the manufactured tires, at a cost of 1.78 Euros each. A share of 39% is recycled in retreading and 37% in energy cogeneration. Canada, Japan, and USA disclose that they recycle 89% of the tires produced by means of diversified processes, mainly in energy co-generation systems [5,15,16]. In the 2000s, Korea and Spain also adheredto pyrolysistechnologyforused tires [16,17].

According to Nourreddine [18], an EU draft directive states a goal that, by 2015, only 5% of a vehicle's weight can be disposed to discharge sites, and that a further 10% can be incinerated. Usually, the recycling of automotive vehicles is focused on recovering metals, while other materials in the form of shredder fluff are disposed to landfills. This material is currently incinerated for energy and carbon black production. The challenge remains because many countries donot have technology to meet the requirements specified in this directive.

The environmental problem caused by used tires is most noticeable in developing countries, as reported by Osahy et al. [4], who mentioned that 160,000 tons of waste tires are generated in South Africa annually, and up to 28 million used tires are dumped unlawfully or burned. Moreover, this figure is estimated to increase by 9.3 million yearly. The authors proposed the elimination of this waste through pyrolysis of used tires to obtain carbon black.

Banar et al. [19] reportedthat, in Turkey, while 8 million tires are produced per year, which is equivalent to 285,000 tons/year, the total installed capacity for recovering waste tires is 101 tons/year. In the same country, Aydin and Ilkiliç [6] were able to obtain a pyrolytic oil thatcan be used as fuel in diesel engines, after removal of excessive sulfur.

In 2012, more than 280 million tires were discarded in China, with a weight of 10.18 million tons. Scrap tires are insoluble and infusible, and are therefore difficult to degrade naturally. A lack of suitable techniques and economic factors over the years has led to scrap tires becoming a serious problem in terms of environmental pollution. At present, most scrap tires aredeposited in open or landfill sites, resulting in disposal problems and the increased risk of fires [3]. Various recycling methods have been developed over the years, such as retreading, incineration, and crumbling to produce rubber powder, but they all have significant limitations or drawbacks [3].

Developed countries have been consistently advised to prevent harmful effects to the environment. One of the options for the management of the scrap tires is material recovery. A prevalent alternative to recover scrap tires is rubber crumb, together with the removal of embedded fiberfor the purpose of new products such asplaygrounds and surfacing solutions [13,20–22]. However, according to Banar et al. [19], Williams [22], and Martínz et al. [23], pyrolysis must be regarded asa viable and relevant alternative to recycle the tires because its various derivatives have a greater scope for application and valorization.

*1.2. Pyrolysis as an Ecofriendly Solution to the Scrap Tires*

Pyrolysis of scrap tires has evolved as a viable alternative to overcome the practices of incorrect disposal of tire waste. Pyrolysis processes can produce tire derived oils that may be used as fuel or added to conventional fuels, producing fuel blends with improved properties at a reduced cost. Pyrolysis is a process that can contribute to overcome the practices of disposal of tire residues from inadequate sites, therefore standing out as a sustainable process to produce alternative fuels [15].

Roy et al. [5] defined the vacuum pyrolysis as the combustion of organic substances in the absence of air. It enables the production of large quantities of pyrolysis oils from organic substances. Vacuum minimizes secondary reactions such as thermal cracking, repolymerization and recondensation reactions, gas phase collision, catalytic cracking and redox reactions. If the vapor phase products are quenched, the yield of organic liquids such as pyrolysis oils is increased at the expense of solid residues and gases. The physicochemical properties of the end-products are a function of the pyrolytic temperature [24]. According to Al-Lal et al. [25], Martinez et al. [23] and Raclavská et al. [26], the pyrolysis processcanbe used to obtain fuel from biomass, coal, lube oil, plastic and tire wastes.

In the last three decades, many products have been derived, thus affirming the interest of pyrolysis for waste tires. For example, Yousefi et al. [27] developed polymer-modified asphalts prepared by incorporating recycled polyethylene and a used-tire-derived pyrolytic oil residue in asphalt. Its characteristics showed superior properties for the modified asphalts at high temperatures.

At low temperatures, the bitumen becomes brittle and cracks, while, at high temperatures, it softens with the result that the bitumen binder either migrates to the surface or the pavement tends to be put under stress. To solve these problems, Chaala et al. [28] mixed carbon black with bitumen at levels between 5% and 30%, reaching significant results on the rheological behavior of asphalt obtained.

Canada ranks among the first countries touse tires as a source of energy and of valuable chemical products by thermal decomposition of rubber in a pilot plant. Yields are 55% oil, 25% carbon black, 9% steel, 5% fiber and 6% gas. The maximum recovery of oil isperformed at 415 °C below 2 kPa. The energy obtained from tire pyrolysis has been estimated in 700 kJ/kg, with a mass flow of 200 kg/h [29].

However, according to Roy et al. [5], the pyrolysis of scrap tires has been, thusfar, uneconomical due to the absence of an established market for oil and, in particular, for the pyrolytic carbon black product. Therefore, most existing research ontire pyrolysis is more concerned with obtaining pyrolytic oil, which can be used in diesel engines. Several authors showed that the application of pyrolytic oil in engines is feasible, because it hasa quality equivalent to diesel oil [4,6,30–36]. Mui et al. [33] also

showed that it is possible to use carbon black in the treatment of textile industries effluents. Debek and Walendziewski [35] showed that, after hydrorefining oil from tire pyrolysis, it is possible to obtain good quality fuels that can be used in passenger carsand vans.

Furthermore, in a survey addressing environmental impacts, Huijbregts et al. [36] calculated product-specific ecological footprints from consistent and quality-controlled life cycle information of 2630 products and services, including energy, materials, transport, waste treatment and infrastructural processes. They showed that the disposition of tires to a waste incineration process has an ecological footprint of 72 $m^2$/year, while its disposal in the landfills hasan ecological footprint equal to 113 $m^2$/year, that is, its impact is reduced by almost 40% when incinerated. In this way, it is evident that the incineration of tires is a greener way than disposal in dumps or landfills.

Currently, this technique ranks among the best to mitigate the contamination from tires discarded inadequately and consequently it concedes a better end-life to the tires [25,28–41].

According to Umeki et al. [24], the pyrolysis oil of the scrap tires has black coloration, strong odor and specific gravity around 0.93 $g/cm^3$. In addition, the compositional analysis of tire pyrolytic oil (TPO) describes the liquid as a complex mixture, composed mainly of aromatic compounds and olefins, containing important fuels such as gasoline and diesels. They concluded that the fuel blend's properties point to a potential viability of using the TPO in mixture with diesel. This would be an alternative fuel for automotive and industrial uses and for the replacement of conventional petroleum fuels.

The energy efficiency of the diesel oil blend with the tire pyrolysis oil was tested on a 440 $cm^3$ single-cylinder diesel engine. Engine performance, evaluated at different engine speed and loads, showed that the use of 20% of weight (%wt) blend does not cause significant differences in terms of torque, power, specific fuel consumption, and exhaust emissions, compared to those obtained using diesel fuel [40]. Similar results were obtained by Wang et al. [2] whotested several diesel oil blends with pyrolysis oil from waste tires on a diesel engine for rotations up to 2500 rpm.

In Iran, Hossain et al. [41] produced a pyrolysis oil from a mixture of scrap tire and rice husk with characteristics very close to petroleum oil. The highest fuel oil yield was 52 wt% when a mixture of 50 wt% tire and 50% rice husks was used, which was pyrolyzed at 450 °C. The results show that it is possible to obtain liquid products comparable to petroleum fuels, and valuable chemical feedstock from the selected wastes if the pyrolysis conditions are chosen according to the products to be obtained.

Ayanoglu and Yumrutas [11] producedgasoline- and diesel-like fuels from waste tire oil by using catalytic pyrolysis in a heat reactor. After the distillation processes, the fractions obtained were composed of 18 wt% of light oil (gasoline), 70 wt% of heavy oil (diesel fuel) and 12 wt% of residues. Furthermore, the carbon distribution of GLF ($C_4$-$C_{12}$) and DLF ($C_{13}$-$C_{17}$) samples was close to the one of standard fuels.

Li et al. [3] developed a continuous process of pyrolysis from the scrap in the presence and absence of catalysts. The maximum yield of derived oil was up to 55.65 wt% at the optimum temperature of 500 °C. The catalytic pyrolysis was performed using 1.0 wt% (on a scrap tire weight basis) of catalysts. They concluded that the derived oil can therefore be used as a petrochemical feedstock for producing high-value-added chemical products or as fuel oil.

Ahoor and Zandi-Atashbar [1] obtained pyrolysis oil under an argon atmosphere at 407.3 °C. They achieved 12 wt% of fuel oil, the highest yield. Several works have shown that the derived oil contains variable concentrations of valuable aromatic and aliphatic compounds such as butadiene, D-limonene, benzene, toluene, and xylenes, which could be used directly as substitutes for conventional fuels or petrochemical feedstocks as a potential source of light aromatics [3,42].

These tire pyrolysis plants are not yet widespread in underdeveloped countries because of their high cost of deployment. However, researchers have dedicated efforts to enable its implementation as follows.

In Turkey, Ayanoglu and Yumrutas [11] developed a low-cost tire pyrolysis plant. The main part of pyrolysis unit cost was US$11,477. The main part can be used for 10 years with a full load

production. Amortization of the pyrolysis unit was 0.157 US$/L, and they realized that the cost of production of the other derivatives was above the price paid for oil products in Turkey.

To minimize the cost of production, Luo and Feng [43] used the waste heat of blast-furnace slag in the production of fuel oil and combustible gas by catalytic pyrolysis, as a novel waste energy recycling strategy. Their results show that there was an upgrade in the quality of pyrolysis oil.

*1.3. Brazilian Situation Regarding the Management of Scrap Tires*

According to the National Association of Tire Manufacturers (ANIP), Brazil has 20 companies that manufacture tires, which are responsible for 150,000 jobs [44]. In 2014, the production reached 68.8 million tires. There was a 26% increase in tire production compared to 2006 and, due to imports, 74.9 million tires were sold that same year [44].

According to Machin et al. [10], the production of tires in 2014 by the Brazilian industry totaled 70.8 million units, which was a small reduction compared to 2013, when the sector achieved a historical record. According to the sectorial balance presented by ANIP, Brazil closed 2016 with a fall of 1.1% in total tire production as compared to 2015 [44]. From 2014 to 2015, the accounts also closed in the red, with a decrease of 1.2%. The ANIP balance sheet presented at the end of December 2017 showed a growth of 2.4% in tire sales, as compared to the same period in 2016, which suggests a slight recovery of the sector, following two years of crisis [44].

The Normative Instruction N°001/2010, from the National Council of the Environmental of Brazil (CONAMA), regulates the procedure that manufacturers and importers must meet for registration, calculation of goals and confirmation of the allocation. This law states that 100% of the outstanding tires in the country should be recycled, and determines that the companies are responsible for handling the end of life-cycle and end destination of tires [45].

Manufacturers and tire importers must prepare a management plan to collect, store, and dispose scrap tires within six months after the publication of Resolution No.416/09. The Resolution specifies that, in cities with over 100,000 habitants, at least onecollection point should be installed, within oneyear following the publication of the resolution. The new resolution does not consider the reform of tires as recycling, but as an activity that prolongs tire life [45].

However, in São Paulo, for instance, only fourcollection pointswhere created when the regulation proposed the creation of 120. In addition, there is no report about the volume of discarded tires in landfills [16]. Currently, there is no register about the collection points in other cities from Brazil. This demonstrates a disregard of companies towards Brazilian politics, that probably can be explained by the fact that companies want to avoid the costs associated with the collection and implementation of ecologically friendly processes for the end destination of the tires [5].

According to ANIP in 2016, the goal established by CONAMA was reached and the manufacturers of tires were able to give a correct destination to 404,328.13 tons of waste tires and, from January to September 2017, over 360,000 tons of tires had the correct destination, which corresponds to more than 4.0 million tons of waste tires collected since 2009 [44]. However, according to Lagaritos and Tenório [16], Brazil only repairs half of the waste tires at a cost of US$0.45/tire, which can be proven by the tires in dumps, rivers and lakes that are constantly caught by public inspection entities.

However, according to Roy et al. [29], all of the tire recycling and treatment processes cited above have some disadvantages. Retreading can only be performed when the carcass is not damaged. When tires are used as solid fuel, polycyclic aromatic hydrocarbons and soot are produced. Therefore, expensive gas cleaning devices are necessary for the removal of potentially hazardous compounds. Tire grinding is very expensive since it is performed at cryogenic temperatures or requires energy-intensive mechanical equipment.

A tire recycling technique, used in many countries, but not yet extensively used in Brazil, is the tire pyrolysis. The outputs of this processing alternative can be used in the preparation of dyes for paints and varnishes, as blends for rubber or asphalt and obtaining some chemicals such as limonene and fuel additives [27,28,46,47].

As noted, tire pyrolysis is a feasible and ecologically friendly alternative as the end destination of tires [38], since it produces energy and various chemical products used as fuels (including gasoline), dyes, polymers, asphalts, etc.

This paper describes a case study in Brazil, which illustrates how to reduce the environmental impacts caused by tire disposal in a sanitary landfill alongside making the process profitable, by vacuum pyrolysis. Moreover, this work aspires to contribute to the development of a market for the pyrolytic oil and carbon black products obtained from a vacuum pyrolysis process, in Brazil. Following a preliminary process of vacuum pyrolysis of used tires conducted in laboratory, the process has been scaled up over the last 10 years from a batch to pilot plant with 100 kg/h capacity.

## 2. Materials and Methods

### 2.1. Layout of Tire Pyrolysis Plant

Figure 1 shows a scheme of the tire pyrolysis tire where this work was developed. The recycling cycle was composed by a sector tire reception, followed by a sector for storage while pre-heating them at 100–110 °C. All processing conditions were based on Chaala et al. [28] and Roy et al. [5]. The heat was generated by a furnace that fires fuel oils. Tire samples were loaded by a conveyor wagon that entered them into the reactor for putting the samples in contact with internal heat. The reactor has a capacity of heating two wagon at the same time. Gaseous products from the reactor were condensed in a distillation tower for separation of gas, light oil, heavy oil, and crude oil, whichare used as fuel, polymers and asphalt components. The solid products from reactor were cooled in a heat exchanger, after which a magnetic separation to remove metalwas applied. Afterwards, they were crushed in a blade mill and mashed to obtain carbon black, which is sold to dyeing companies. However, to obtain the experimental data, a scale up of the reactor was fed with a mass flow of 25 kg/h of tires at 600 ± 50 °C and 20 kPa. The reactor temperature was monitored on line by an internal thermostat. Samples were collected periodically for monitoring the pyrolysis of tires.

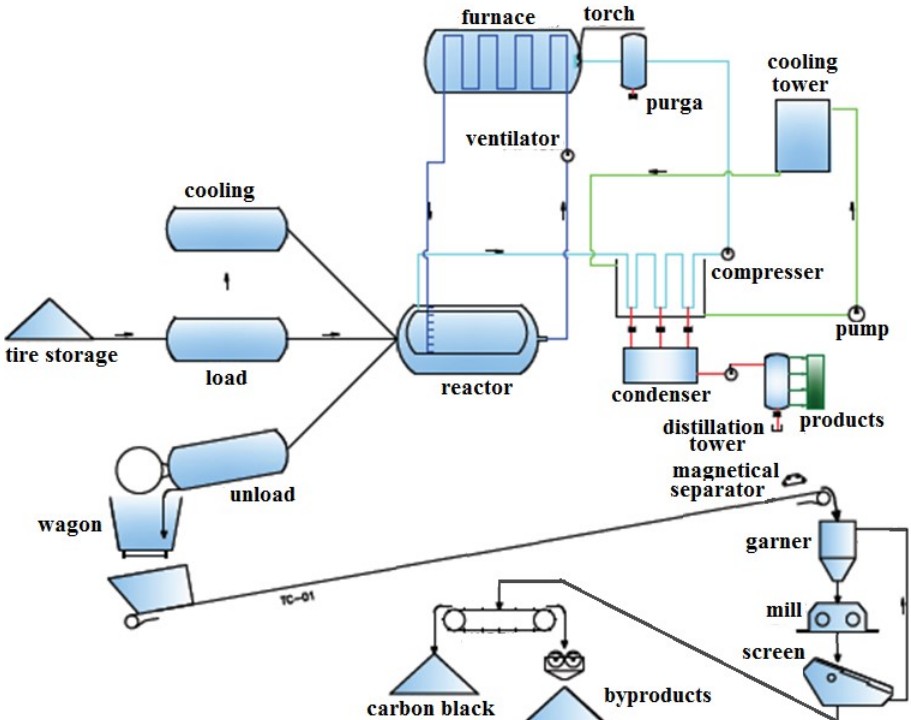

**Figure 1.** The used tire pyrocycling process flow sheet.

### 2.2. Chemical Analysis of Products

Characterization of liquid product: The chemical composition was analyzed by gas chromatography coupled with mass spectroscopy (GC-MS), near region (NMR) and Fourier transform infrared spectroscopy (FT-IR) and standard methods of the American Society for Testing and Materials (ASTM). The inferior calorific value was measured by the ASTM D129 method. The flash points of oils were determined by the Dean and Stark method. The value of carbon Conradson residue was obtained by ASTM D524. The ketone index, the sulfur content and density were measuredby the ASTM D4737, ASTM D3177 and ASTM D4052 methods, respectively [48]. The elementary composition was determined with a LECO CHN-600 apparatus [30,34,35,38,49].

Characterization of Carbon black: This product wascharacterized by nitrogen adsorption, electron spectroscopy for chemical analysis (ESCA) and inverse chromatography [49].

A shelf life study of pyrolytic oil was done to assess the possibility of using oil as a fuel additive. To this end, pyrolytic oil was mixed at 2% and 3% with diesel oil and samples were storedfor 21 days in dark conditions. Afterwards, all ANP parameters were measured in samples [48] asrequired by Brazilian laws.

### 2.3. Price Description and Cost Analysis

Former records registered the investments in plant construction aroundUS$ 1.25 million, for an estimated 10-year lifetime. The cost for plant maintenance (depreciation) was 4% of the total investment, thusthe annual fixed costs totaledUS$130,000. The warehouse rate is a revenue source and thusUS$0.45/tire will be charged tothe tire companies, in accordance with Lagarinos and Tenório [16], a price already practiced in Brazilian market. According to MFRural [50], carbon black is sold in Brazil at US$1.50/kg and the fuel additive is sold in Brazil at 3.50 US$/L. According to SEFAZ [51], steel scrap is sold for0.03 US$/kg and, according to SINDGAS [52], the sale price of the cylinder containing 13 kg GLP gas was US$19.36 inSeptember 2017, in Brazil. The average costof energy in the region of the company is 0.13 US$/kWh [53]. The calculation was based on maximum capacity of pyrolytic reactor of 876 ton/year (100 kg/h), considering its operation in continuous flow feeding by skip cars. This way, the calculations was based on annual mass processed by the company. All costs and profits are summarized in Tables 6–8. All calculation procedures were presented by Ayanoglu and Yumrutas [1]; Almeida et al. [54]; Benvenga et al., [55]; Giraçol et al. [56] and Passarini et al. [57].

### 3. Results and Discussion

### 3.1. Performance of the Pyrolysis Process

Figure 2 shows the variations of temperatures for the combustion gas (outside), of the inner wall of the reactor and the reaction temperatures of the materials measured during the pyrolysis process. Temperature stabilization occurredafter 40 min of process time due to heat transfer between the environments involved until the thermal equilibrium of the reacting systemwas reached. This time can be considered as the beginning of the standing state of tire pyrolysis process. Finally, there was a drop in temperature due to disrupting the power supply of the system, as this is a batch process. A summary of the thermal analysis is shown in Table 1.

In this continuous process, the reaction heat isused forpreheating the next sample of tire, which are loaded into the reactor by conveyor wagons and the initial 40 minof the process timeis eliminated, thus reducing the overall production time [2,29].

To determine the proportion of the liquid formed, and the solid and gaseous products, the masses of each component were measured. Table 1 shows the mass balance obtained with three assays for the pyrolysis of crushed tire samples. As noted, the time for the complete reaction was 1.43 h and the major components of pyrolysis productswere 41% of pyrolysis oil and 38% carbon black.

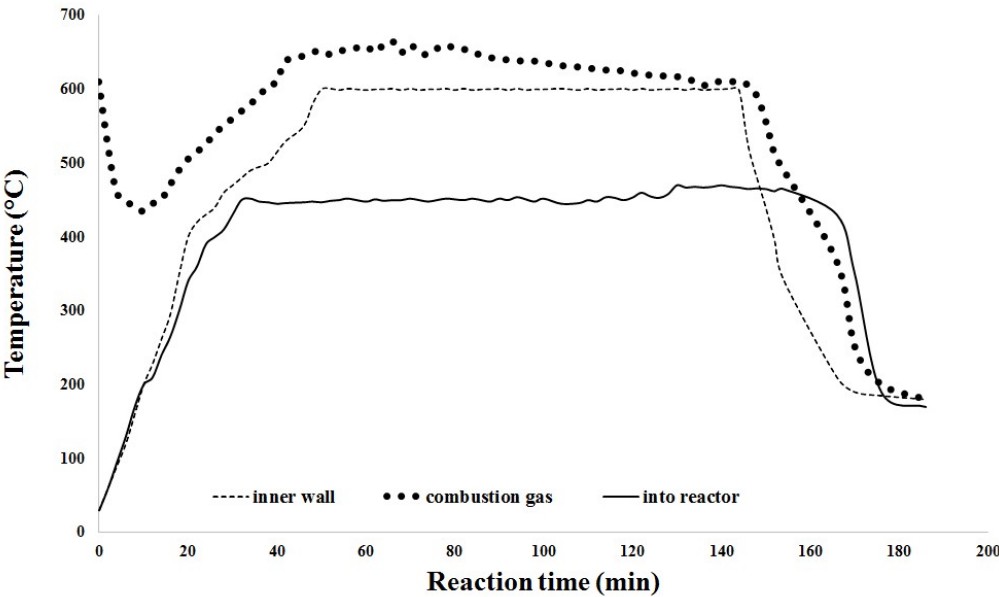

**Figure 2.** Variation of temperatures during the tire pyrolysis process.

**Table 1.** Mass balance of tire pyrolysis process at 600 °C and 20 kPa.

| Sample | Carbon Black (%) | Pyrolytic Oil (%) | Pyrolytic Gas (%) | Stell Scrap (%) | Reaction Time (h) |
|---|---|---|---|---|---|
| Tires | 38.35± 0.25 | 40.72± 4.16 | 12.12 ± 4.32 | 8.95± 0.05 | 1.43 |
| Calorific value (kJ/kg) | | | | 34,842 ± 4,174 | |

Table 2 shows the chemical composition of the gaseous fraction samples that passed in the distillation column. As noted, the major component of the gaseous fraction are fuel gases composed of48.85% natural gas (methane and ethane), 26.45% gas liquefied under pressure (GLP) (propane and butane), and 24.70% $CO_2$ derived from tire combustion. The liquid fraction obtained in this work was light oil of 0.95 specific gravity and dark color, composed of66% gasoline, 10% hexanes that can be used as an additive to gasoline (limonene) [6,29,48] and 24% high oils that can be used as fuel for blast ovens [48].

Tables 1 and 2 show the analysis of lower calorific power and gases obtained from tire pyrolysis. The calorific values in this study were: 36 MJ/kg for tire, 42 MJ/kg for GLP and 60 MJ/kg for natural gas. These values can be regarded as good since it is a mixture of natural gas and GLP (methane, ethane, propane and butane). These products can be used as an energy source for heat generation and, therefore, can be negotiated by the company as further products generated in the process.

In a similar research work, Roy et al. [29] obtained yields around 55% oil, 25% carbon black, 9% steel, 5% fiber and 6% gas. The specific gravity of this oil was 0.95, its gross heating value was 43 MJ/kg and total sulfur content about 0.8%. It was rich in benzol and other petrochemical components. This oil is suitable for mixing with asphalt [27,28].

The yield of the oil was higher than those obtained by Ahoor and Zandi-Atashbar [1] and Ayanoglu and Yumrutas [11], but smaller than those obtained by Li et al. [3] and Hossain et al. [40].

However, according to Alkhatib et al. [40], the oil composition depends on the heat received by the scrap tire, and in the best condition they obtained 45% oil. It was observed that increasing the heating power supplied to the pyrolysis reactor from 750 to 1500W resulted in an 11% increase of oil produced; however, the tar was too high.

**Table 2.** Results of chromatographic analysis gaseous fraction from tire pyrolysis to show its chemical composition.

| Substances | Mass Composition (g/g, %) |
|---|---|
| Carbondioxide ($CO_2$) | 24.70 ± 0.23 |
| Metane ($CH_4$) | 36.77 ± 4.15 |
| Etane ($C_2H_6$) | 12.09 ± 1.38 |
| Propane ($C_3H_8$) | 9.19 ± 2.05 |
| Isobutane ($C_4H_{10}$) | 1.94 ± 0.20 |
| 2-Metilpropane ($C_4H_{10}$) | 6.57 ± 0.67 |
| 2-Metilpropene ($C_4H_{10}$) | 6.30 ± 1.38 |
| n-Butane ($C_4H_{10}$) | 2.44 ± 0.20 |
| Natural gas calorific value (kJ/kg) | 42,420 ±3089 |
| GLP calorific value (kJ/kg) | 60,128 ±1256 |

Novicki and Martignoni [58] statedtheir tire calorific value was 30.2 MJ/kg, while the calorific value of their gas was 33.6 MJ/kg. Thus, the tire pyrolysis and GLP calorific values obtained in this work were higher than those ofNovicki and Martignoni [58].

However, other authors achieved higher values for energy generated by tire pyrolysis, e.g. Banar et al. [19] obtained a calorific value of 37.5 MJ/kg and Oliveira et al. [59] measured the calorific value at 42.0 MJ/kg.

*3.2. Pyrolytic Oil as an Additive to Fuel*

Table 3 shows the chemical composition of samples of the liquid fraction retained in the distillation column by gas chromatography. The liquid fraction obtained in this work was light oil of 0.95 specific gravity and dark color, which was composed of 66% gasoline, 10% hexanes that can be used as an additive to gasoline (limonene) [28,47] and 24% high oils that can be used as fuel for blast ovens [31].

As noted, the average molecular weight of the blend of oils is low, indicating it as an additive for other fuels. According to Santos et al. [60,61], even if the oil has a high molecular weight, it would be an optimal fuel in blast furnaces or cogeneration as shown by Banar et al. [19] and Santos et al. [60,61].

According to Santos et al. [60,61], oil companies have expressed interest in unconventional oil as alternative resources for energy supply, mainly because an increase of 40% in world market energy consumption is forecasted for 2035, and this would facilitate the selling of oil produced from pyrolysis of used tires.

Garcías-Contreras et al. [62] developed a residential boiler of 29.1 kW using a tire pyrolysis liquid (TPL)/diesel fuel blend (50/50 vol.%), as an alternative to heat in households. When they compared the exhaust temperature of diesel combustion gases with the TPL blend, there were no differences and both fuels tested increased the water temperature similarly. This demonstrates the feasibility of using tire pyrolysis oil in boiler. Wang et al. [2] and Martinez et al. [23] also observed the similarities of tire pyrolysis oil andpetroleum. However, Frigo et al. [40] noted that in vitro assays on particulate emission for diesel and TPL/diesel blend had similar cytotoxic potency and no genotoxic effect.

Table 4 shows the results of fluid analyses of quality of diesel oil mixed to pyrolytic oil, ascompared with the ANP standards [48].

The liquid obtained by condensation of the reaction vapors showed a dark aspect, probably due to the presence of fine particulate matter in suspension (carbon). To test the liquid fraction as an additive in fuel, it was mixed at 2% or 3% with diesel oil. Table 4 shows the analysis of diesel oil additive after 21 days of storage. For a better understanding of the values, Table 4 also includes the limits required by National Agency of Petroleum [48] of Brazil for distillation temperature and the products obtained from petroleum decomposition process.

**Table 3.** Of chromatographic analysis of liquid fraction (oil) from tire pyrolysis to show its chemical composition.

| Substances | Mass Percent (g/g, %) |
|---|---|
| Isopentane ($C_5H_{12}$) | $1.78 \pm 0.08$ |
| 1-Pentene ($C_5H_{10}$) | $1.07 \pm 0.08$ |
| 2-Pentene ($C_5H_{10}$) | $7.00 \pm 0.12$ |
| 1,3-Pentadiene ($C_5H_8$) | $36.47 \pm 0.67$ |
| 2-Metil-1butene ($C_5H_{10}$) | $0.32 \pm 0.08$ |
| 2-Metil-2butene ($C_5H_{10}$) | $17.84 \pm 0.67$ |
| n-Pentane ($C_5H_{12}$) | $1.38 \pm 0.08$ |
| Hexanes(~C6) | $10.09 \pm 4.15$ |
| Higher oils | $24.05 \pm 0.67$ |

This analysis aimed to determine the shelf life of the diesel oils by simulating storage conditions of the fuel and checking the formation of turbidity and precipitation of materials in the liquid. This test showed that all samples studiedat2% mix remained within this parameter. All 3% samples showed deposition material and some turbidity, thus did not meet ANP specifications [48].

**Table 4.** Shelf life study of thediesel oiladditive.

| Characteristic | Control | Pyrolyticoil | | ANP Standard |
|---|---|---|---|---|
| | Dieseloil | 2% | 3% | |
| Aspect | Clear | Clear | Clear | Clear |
| Cor | 2 | 2 | Black | 3 |
| Boil point (°C) | 132 | 132 | 131 | Note |
| Temperature (50% evaporate) | 274 | 274 | 288 | 245 to 310 |
| Temperature (85%evaporate) | 340 | 340 | 346 | 370 |
| Evaporation point (°C) | 382 | 382 | 392 | Note |
| Specificmass ($Kg/m^3$) | 854 | 854 | 874 | 220 to 880 |
| ketone index | 0 | 0 | 0 | 45 |
| Sulfur (mg/Kg) | 400 | 400 | 1150 | 1800 |
| Flash point (°C) | 51 | 51 | 57 | 38 |

Trying to encourage the recycling of tires, Banar et al. [19] obtained a pyrolytic oil of high molecular weight from pyrolysis of used tires in Turkey, which canbe used for energy cogeneration according to the country laws. The oil obtained by Chaala et al. [28] had a specific gravity of 0.95, gross heating value of43 M J/kg and total sulfur content of about 0.8%. It was rich in benzol and other petrochemical components, i.e. it was also a heavy oil. However, in this work, pyrolytic oil was mixed with fuel oil to show it can used as fuel for furnaces, but it also can be used as motor fuel, according to Brazilian laws [47].

Ahoor and Zandi-Atashbar [1], Ayanoglu and Yumrutas [11], Hossain et al. [41], and Umeki et al. [24] obtained an oil of similar quality to that presented in this work, reporting that the oil had properties similar to a gasoline or a petroleum oil. Frigo et al. [43] and Wang et al. [2] demonstrated the energy efficiency of this oil in the production of energy in a diesel engine. Moreover, this viability of pyrolytic oil used in engines wasalready confirmed by other authors [4,6,30–36,38,58].

*3.3. Characterization of the Solid Fraction*

The solid fraction, which is inside the reactor after the pyrolysis process, is scrap metal and/or coal. The coal, depending on the reaction temperature, can be milled and sold as semi-reinforcing filler or filler for the rubber and paint industries, and sold as dyes or coal for use as fuel. In the case of metals contained in tires, they may be sold as scrap tometallurgic companies. To determine the proportion of carbon, nitrogen, hydrogen and sulfur, elemental composition measurements were performed aiming at possible application of these materials in the steel industry as a source of "fine" for blast furnace base

injection. Thus, the parameters Fixed Carbon (FC), volatile and ash were determined. These variables are chemical parameters of reference for the quality of coal fines for steel.

Table 5 shows thethermogravimetricanalysis results for solid fraction in an oxygen atmosphere. In this analysis, the carbon was oxidized at 400–600 °C, whereby the organic material was transformed into pure carbon. The carbon black was 3% volatiles, 9% ashes, and 88% fixed carbon, of which 86% was pure carbon. According to Roy et al. [29], this composition ensures its use as raw material for dyeing and inks industries, in addition to other applications cited throughout the text. Sulfur is not a desirable component in carbon black, and the sulfur content (2%) obtained in this study was consistent with most research work reported [5,19,28].

Osayi et al. [4] conducted a review and found ranges of 74–86% for C, 5.8–7.5% for H, 0.2–1.8% for N, 1.0–2.1% for S and 2.1–14% for ashes. However, Banar et al. [19] cited a composition with 82.5% carbon, 6.9% hydrogen, 8.4% oxygen, 1.7 sulfur and 0.5% nitrogen, for the carbon black obtained in their work. Therefore, the carbon content of the product obtained in this study was better than those cited by Banar et al. [19] and Osayi et al. [4].

Novicki and Martignoni [58] stated that a carbon black ideal to steel industry must have more than 75% carbon content and less than 10% volatile content. Moreover, Banar et al. [19] obtained carbon black with 24.1% fixed carbon, 65.5% volatiles, 9.63% ashes and 0.84% moisture from tire pyrolysis. Chaala et al. [28] broughta carbon blackto theCanadian market with 77% pure carbon,19% ash and 4%volatiles. As can be seen, thecarbon black obtained in this study hasa better quality than that of Banar et al. [19] as well asthose required by the Brazilian steel industries and Canadian market.

**Table 5.** Evaluation of carbon black obtained in this work.

| Source | Percent composition (%) | | | Elemental composition (%) | | | | |
|---|---|---|---|---|---|---|---|---|
| | Volatile | Ash | FC | C | H | N | S | Other |
| **Carbon Black** | 3.25 | 9.05 | 87.7 | 86.49 | 1.30 | 0.51 | 1.96 | 9.74 |
| **Deviation** | 0.55 | 2.05 | 2.60 | 2.70 | 0.01 | 0.18 | 0.13 | 2.38 |

OBS: FC, fixed carbon.

*3.4. Environmental Cost Accounting of Pyrolysis Process of Tires*

Tables 6–8 show the profitability of 876 ton/year (100 kg/h) plant for vacuum pyrolysis of used tires. Because the pyrolysis process aims to be auto-sufficient in energy, the fuel oil used in the startup process was not considered in this calculation, because its costs couldbe addressed as relatively insignificant as they need only one utilization over the whole operation period.

**Table 6.** Calculation of expenses of process.

| Type of Expenditure | Quantity | Annual Cost (US$/year) |
|---|---|---|
| Financial expense | | 124800.00 |
| Taxes on sale (%) | 18 | 346,923.39 |
| Depreciation (%) | 4 | 5000.00 |
| Cost employees | | |
| Engineer | 1 | 18000.00 |
| Technicians | 9 | 75600.00 |
| Administrative offices | 2 | 8400.00 |
| Taxes on salaries (%) | 36 | 36720.00 |
| **Sum cost employees** | | **138,720.00** |
| **Total expenses (US$/year)** | | **615,443.39** |

As shown in Table 6, the higher process costs were associated with the commercial taxes (55%) due to product sale, salaries (23%) and the financing of the plant (21%). Even in this scenario, the costs amounted to one third of the revenues.

Table 7 displays the revenue sources for process the tire pyrolysis. It shows that the main products influencing the revenues (87.6%) were the pyrolytic oil and the carbon black, at 61.5% and 26.1%, respectively, reaching a total revenue of US$ 1.927 million and a total profit US$ 1.312 million. It is also noticeable that tire companies need to pay rights concerning the tire disposals, according to the Brazilian law, thusthe pyrolytic plant has an additional revenue of more than US$78,000 per year for tire storage [16].

For a pyrolytic plant of 200 kg/h of tire, Roy et al. [29] demonstrated that the process feasibility is promising, with returns on investment of 31% after three years of operation, in Canada. Based on this and the data inTable 6, all the company's financing costs can be covered by the total profit of the first year of operation alone. In this way, the plan proposed by this work proved to be three times more economically viable than the one described by Roy et al. [29].

**Table 7.** Calculation of the revenues of the tire pyrolysis.

| Product | Quantity | | Price (US$/Unit) | Individual Revenue (US$/year) |
|---|---|---|---|---|
| | **Hourly (kg/h)** | **Annual (kg/year)** | | |
| Pyrolytic Oil ($m^3$) | 40.68 | 338,539 | 3500.00 | 1.184,886.40 |
| Carbon Black (ton) | 38.31 | 335,596 | 1500.00 | 50,3393.40 |
| Steel scrap (ton) | 8.9 | 77,964 | 30.00 | 2,338.92 |
| GLP (13 kg) | 12.1 | 8,154 | 19.36 | 157,893.44 |
| Tire storage (unit) | 20 | 175,200 | 0.45 | 78,840.00 |
| Revenues (US$/year) | | | | **1,927,325.16** |
| Total Profit (US$/year) | | | | **1,311,908.77** |

Table 8 shows other possible gains with unusual by products such as carbon credit and energy sale from cogeneration. As can be seen, more than 1600 carbon credits can be claimed after certification by governmental certifying agency and in addition to the US$15,500 the company gains an image of ecofriendly company, which favors the marketing of its products. In addition, it is possible to sell the energy generated during the tire pyrolysis to an energy company, as is the practice in the context of the Brazilian alcohol industry, leading to an additional gain of US$2.445 million per year. Adding these values and subtracting commercial taxes due to product sales, leads to a total profit due to unusual byproducts of US$2.017 million per year.

**Table 8.** With unusual by products per year.

| By emissions | Quantity | Carbon credit (ton $CO_2$) | Price (US$) | Partial profit (US$) |
|---|---|---|---|---|
| Tire (ton) | 876 | 1,016.16 | 9.25 | 9,399.48 |
| Energy (GW) | 26.865 | 658.20 | 9.25 | 6,088.35 |
| Sum | | 1,674.36 | - | 15,487.83 |

| By cogeneration | Quantity | Conversion Efficiency (%) | Price (US$/kW) | Expenses (US$) | Partial profit (US$) |
|---|---|---|---|---|---|
| Energy (GW) | 26.865 | 70 | 0.13 | | 2,444,730.00 |
| Taxes on sale (%) | 18 | | | 442,839.30 | |
| Profit of unusual byproducts (US$/year) | | | | | **2,017,379.00** |
| Total profit + unusual profit (US$/year) | | | | | **3,329,287.77** |

Overall it is possible to reach a total profit of US$3.329 million per year. If the company wishes to pay completely the funding, its profit would be US$2.204 million in the first year and US$3.454 million

in the following years. This project is more economically viable than projects described by Ayanoglu and Yumrutas [11] and Roy et al. [5].

### 3.5. Advantages of Tire Pyrolysis

Tire pyrolysis is energetically self-sufficienct, given that the energy required for pyrolysis comes from the process itself [38,63,64]. The process promotes tire recycling, namely the rubbersand metals contained in these materials. Moreover, it is a clean production process, because there is no usable waste recycling so it does not generate environmental liabilities [36,64].

The process has commercial viability since the sub-products generated have a production cost lower than the market prices [11,36,63,64]. Other advantages include the fact that the process is easy to operate and maintain, with low costs [11]; it does not generate odors [16,64]; and it is an innovative and environmentally friendly solution for old tires [63,64].

There will be revenue upon receipt of tires or rubber products, thusthere is no cost for raw materials, because the Brazilian company manufacturing and marketing tires are required to give an environmentally friendly end tires; thus, it can charge fees for receiving the tires from these companies [16,43].

An additional benefit is the possibility of generating carbon credits [55–57] and increase local employmentas well as financial transactions [36,63,64], adding to the social sustainability arguments.

The pyrolysis process can eliminate problems related to several issues, including the needs for space for tire storage, anddifficulties in compression, in transport, and in handling. Tires buried in soil or sunkin water tend to rise to the surface, and stacked tires serve as an ecosystem to rodents and insects that are disease transmission agents such as dengue, zika virusand yellow fever. When burnt, tires release a highly toxic waste in thesoil and increaseair pollution. Moreover, a buriedtire hasno foreseeable deadline forits decomposition.

This process is so innovative that there will be no environmental liabilities, fitting in clean production strategies [36].

Moreover, it will produce various raw materials that are used in tires, rubber products, chemical, smelting, recycling and petrol industries, as well as in thermal boilers [5,16,17,38,41].

According to Li et al. [3], pyrolysis as a viable recycling process that has potential advantages in terms of energy recovery and mitigating the disposal problem. The products of pyrolytic degradation of scrap tires could be reused as high-calorific-value gas to meet the energy requirements of processing plants, oil for boiler fuel, or high-value-added chemical feedstocks and the char formed could be used as low-grade activated carbon or carbon black. It has been proved that the derived oil is more suitable for making high-value-added chemicals than for use as fuel, because it contains large amounts of single-ring aromatics [36].

### 3.6. Strategies to Acquire Raw Materials and the Sale of By-Products

Several alternatives can be advanced for the acquisition of raw materials and the sale of byproducts. Raw materialscan be channeled via large distribution networks that includemunicipalities, ANIP, eco-points and companies of retreads or tire sales. As for byproducts, their sale could be supported by marketing strategies aimed at the valorization of the products that result from the recycling process, via severalresellers and distributors, according to the characteristics of the different types of by-products as follows. The metal components are suitable for selling as scrap for industry and steel production [5,16,17]. Carbon black can be used as a load for semi-reinforcing rubber products and dye industry [5,16,17]. Coal can be usedas a raw material or as an energy source for steel companies [5,16,17]. Benzene has applications in chemical industries and laboratories. It is commonly used as an organic solvent andas a raw material for the production of many organic compounds [39,41].

Toluene can be directed to chemical industries and laboratories, as it can be used as an additive in fuels, and as a solvent for paints, coatings, rubber, resins, thinners in nitrocellulose lacquers and adhesives [39,41,47,48].

Xylene alsohas applications in chemical industries and laboratories, being used as solvent and chemical precursor [39,41,47,48]. Limonene (1-methyl-4-isopropenil-cilohexene) similarly can be used in chemical industries and laboratories [39,41,47,48], with application as a biodegradable solvent and additive of fuels (up to 3% concentration). According to Danon et al. [65], at least 2.5 wt% of a steel-free tire can be converted to di-pentene.

Gas type GLP can be used as an energy source for feeding the pyrolysis process, whereas the excess could be stored in gas cylinders for sale [11,42,63]. In addition, carbon credits, whichexist in theory, could be obtained [55–57].

An additional source of profit could involve the application of a price to the collection of tires from tire resale companiesto eliminate their environmental liabilities [36,66,67]. According to Roy et al. [5], in USA, fees are charged to dispose tires, for which a value of US$1/tire is estimated, while, according to Lagarinos and Tenorio [16], in Brazil, this fee is US$ 0.45 /tire.

## 4. Conclusions

In the conditions described in this study, the yield of pyrolysis process was 41% pyrolytic oil, 38% carbon black, 12% gas and 8.9% steel scrap, with a calorific value of 36 MJ/kg of tire.The carbon black was composed of 90% carbon, which has higher quality than required by the steel and ink industries, and the pyrolytic oil was composed of 66% gasoline and 33% other oils, with sufficient quality to be used as an additive for fuels, or as fuel for engines and furnaces.The analysis showed that the resulting products, pyrolytic oil and carbon black, were within the standards established by Brazilian law and were similar or better than reported in the literature, enabling their use invarious industries.Moreover, it wouldturn usedproducts into raw materials that can be used invarious industries. The main products were carbon black and pyrolytic oil.

Revenue obtained from the sale of products generated by pyrolysis of tires was US$ 1.927 million and the total profit was US$ 1.312 million. It is noteworthy that the total profit could fully pay the costs to the company in the first year and that the total profit couldbe US$3.455 million carbon credits and energy were sold.

Thus, the environmental benefits of pyrolysis process of scrap tires include avoiding the disposal of tires directly into landfills and wasteland, thus avoiding the contamination of the soil, water, and air due to emission of chemical contaminants formed during the decomposition of the scrap tires. In addition, there is no need for the extraction of raw materials for the production of steel, fuel and carbon black produced in the scrap tire pyrolysis plant. The steel produced in the scrap tire pyrolysis plant is cleaner than the steel produced in the current modes. The carbon black produced in the pyrolysis process of scrap tires is cleaner than the currently marketed carbon black. The proposed method produces cleaner fuels than petroleum byproducts, and, as the pyrolysis process generates many products, the fuels have low cost. The use of gas and fuel oil from the tire pyrolysis process will supply part of the Brazilian population's need for these fuels. The recycling of scrap tires into fuel production may reduce the search for new oil fields. Other benefits result from the fact that, as the decomposition of the tires is via vacuum pyrolysis, there is no consumption of air, biotic or abiotic materials, and, as water is used only in refrigeration, it can be considered that water consumption is negligible. Since the plant feeds itself on energy, it does not influence the consumption of renewable sources to supply the needs of its production (the Brazilian energetic matrix is essentially from renewable sources). As all gases, liquids, and solids are not emitted into the environment, there will soon be no contamination of soil, air, and water derived from the plant of scrap tire pyrolysis process, and, consequently, the population neighboring the plant will have a good image of the company. By avoiding the release of greenhouse gases, the company will be a creditor of carbon credits, which will make the company environmentally friendly, and ultimately increase its client portfolio.

Other relevant findings arerelated tothe social benefits of pyrolysis process of scrap tires as follows: the collection of scrap tires for pyrolysis will reduce the volume of garbage that will be carried to the dumps and, consequently their costs, which may then be invested in other benefits to society.

In addition, the negative effects of scrap tire clusters on cities will be avoided. Pyrolysis plants do not release toxic or greenhouse gases, and thuswill not be responsible for the generation of respiratory diseases or climate change in cities of the region. As the plant cogenerates energy, the energy excess will help to supply the energy demand of surrounding cities. The generation of direct and indirect jobs in the city and region surrounding the tire pyrolysis plant will also contribute to the increase of the purchasing power of the people and consequently allow access to food, housing, and better healthcare and education. The increase in the flow of capital in the region will lead to an increase in trade in goods and services;with the increase of commerce, there will be an increase in tax collection and, therefore, municipalities may invest in improvements in public spaces, schools and hospitals in cities.The cities can also concentrate some of the taxes on the improvements of water distribution networks, and on the collection and treatment of sewage, thus improving their image and avoiding polluting the environment due to the scrap tire discard. With the increase in the purchasing power of individuals, the company will be contributing to the improvement in the quality of life of the residents of the surrounding cities; as the pyrolysis plant is a creditor of carbon, its products are derived from the recycling of a waste, soon it has a good image before society, as an ecofriendly company.

Thus, the company can gain with energy sale from cogeneration, due to the sale of 1600carbon credits and gains an image of ecofriendly company, which favors the marketing for selling its products. Overall, the viability for the proposed alternative for dealing with used tires will depend on the combination of the implementation of an efficient plant and recycling process, with the development of adequate marketing and supply chain strategies that allow for redirecting the resulting recycled products to utilizations that are able to extract value from them, and therefore provide an attractive return.

This work showed that tire pyrolysis is an innovative and ecofriendly process, offering a clean production that is economically viable andcan be used as a solution for the disposal of used tires in Brazil. As observed in [68], a largest benefit of pyrolysis is its ability to effectively dispose a type of waste that is hard to recycle while offering the possibility of obtaining recycled products that have economic value in several applications. The economic and environmental gains is relevant to promote the adoption of environmental practices [69–71].

**Author Contributions:** Conceptualization, G.C.d.O.N.; L.E.C.C., L.F.R.P., J.C.C.S.; Methodology, G.C.d.O.N.; L.E.C.C.; J.C.C.S.; Formal Analysis, L.E.C.C. and L.F.R.P.; Writing—Original Draft Preparation, G.C.d.O.N. and J.C.C.S.; Writing—Review & Editing, G.C.d.O.N.; L.E.C.C., M.P.C.A.; M.J.F.R. and J.C.C.S.; Supervision, G.C.d.O.N.; L.E.C.C., M.P.C.A.; M.J.F.R. and J.C.C.S.; Project Administration, G.C.d.O.N.; L.E.C.C., M.P.C.A.; M.J.F.R. and J.C.C.S.

**Funding:** This research received no external funding.

**Conflicts of Interest:** The authors declare no conflict of interest.

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
