# Peer review of "Economic, Environmental and Social Benefits of Adoption of Pyrolysis Process of Tires: A Feasible and Ecofriendly Mode to Reduce the Impacts of Scrap Tires in Brazil"

_sustainability, doi:10.3390/su11072076_

Round 1

Reviewer 1 Report

Present paper deals with possibilities of reuse of tires in given region. This topic is highly actual and corresponds with the journal scope. Paper is well addressed, however introduction should be improved. Old tires are currently intensively used as an alternative fuel during cement production; steel reinforcement then serves as source of iron in the clinker. This fact should be mentioned. Lines 261 - 271 are pointless. After minor revision I recommend to publish.

Author Response

Dear Reviewer,

We thank you very much for reading our manuscript and for your contributions that allow us to improve the quality of our writing. Following your comments, we inform that we:

i) conducted a thorough revision of the manuscript, in order to improve the quality and clarity of the writing and the English language;

ii) we revised the introduction for the sake of completeness and included an additional reference, that is very central to the work and was also recommended by another reviewer, notably "Dong, R., & Zhao, M. (2018). Research on the pyrolysis process of crumb tire rubber in waste cooking oil. Renewable energy125, 557-567";

iii) we eliminated the information about the process of selection and extraction of literature references that were in the lines 261-271 from the previous version of the manuscript, as recommended. About this point, we thank the reviewer for the observation. In the version submitted, we had included this information for the sake of clarity and replicability about the process we followed for selecting the papers.

Reviewer 2 Report

the work is appropriate to be published after minor style and grammar editing

Author Response

Dear Reviewer,

We thank you very much for reading our work and for offering us the opportunity to improve it. We took into consideration your observation about the need to conduct some "style and grammar editing" and we did new rounds of reading and correcting the text, checking for typos, grammar errors and other writing issues that were hurting the quality and the flow of the reading. We introduced several corrections throughout all the manuscript.

Reviewer 3 Report

1. General Comment:

This study investigated the benefits of a pyrolysis process of tires in terms of economical and social advantages.

Pyrolysis is a method that realizes thermochemical treatment by breaking chemical bonds of materials under oxygen-free conditions. The largest benefit of this method is that it can effectively dispose wastes that are hard to recycle, and can also get recycled products. At rapid heating rate, waste materials can be converted into higher energy content transportable liquid (such as pyrolysis fuel oil and bio-oil), so pyrolysis is getting more attention. Besides, direct combustion, liquefaction and gasification are other most common approaches for thermochemical conversion of waste (Dong and Zhao, 2018)

Dong R, Zhao M (2018). Renewable Energy, 125: 557-567.

2. Abstract:

2.1. Start abstract with motivation statement. 

2.2. Page 1, Line 16; "This study addresses ..." Use the past sentences! "This study addressed.."

2.3. Page 1, Line 17; "The study describes..." it should be corrected to "The study described.."

2.4. Page 1, Line 21; "Modern economies will increasingly need to devise strategies to address..." Should be corrected to "Modern economies increasingly need to devise strategies to address.."

2.5. Page 1, Line 24; "...this study aims to contribute...." should be corrected to "...this study aimed to contribute...."

2.6. Page 1, Line 28; "Results show yield...." should be corrected to "Results showed yield..."

2.7. Before indicating the results, please mention to the designed system and your methodology in 1-2 sentences!

2.8. Page 1, Line 32; "Novelty or Significance......". Please remove it from the Abstract!!! Try to mention to novelty statement and problem statement in the Introduction!

2.9. Write keywords alphabetically!

3. Introduction:

It is better you bring some data in tables or figures rather than writing in the TEXT.

4. Materials and Methods:

4.1. Page 6, Lines 261 to 271. It is not necessary to mention how finding papers in literature review step! Delete it!

5. Results and Discussions:

I decided to stop reading this paper here!! Beside lack of English, you should compare and discuss several designed systems together to find their economical, social and environmental benefits! Authors just throwed some references without discuss them in details! Discuss mechanism! Compare mechanisms and etc....

Beside improving English, please follow up template of scientific articles!

Author Response

Dear Reviewer,

We thank you very much for the care you took in the revision of our manuscript, and for the opportunity that we were given for improving the quality of the document.

We went trough each of the comments you provided and we addressed them as follows:

Concerning the first general comment, we thank the reviewer for introducing us to the relevant reference in this domain, that we consequently read trough and referred to in the manuscript.

In what concerns the observations for the Abstract, we thank the reviewer for the several issues identified in the abstract regarding the quality of the writing and the relevance of some parts of the text that we chose to include in this section.

In order to address the concern expressed in 2.1, we inserted a sentence clarifying the motivation for the study at the initial part of the abstract: "The work was motivated by the goal of contributing to the development and dissemination of knowledge about existing technologies that allow modern economies to transform waste into valuable products, by documenting and discussing an empirical application in Brazil.".

In order to address 2.2 and 2.3 we changed the writing to the past tense. Following this remark we also went trough the manuscript as a whole and inserted similar corrections concerning the present/past tense of the writing for the sake of coherence and rigor.

The sentence in Page 1, Line 21 was corrected as suggested in the reviewer comment 2.4.

Also following the remarks 2.5 to 2.7 we rewrote the sentences according to the suggestions received.

As for the comment 2.8, concerning having in the Abstract a mention about the "Novelty of Significance" of the work we maintained the writing because of the requirements of the structured abstract, that explicitly asks for the clarification about the Novelty/Significance in this section. So although we understand the comment, we feel like we ought to keep such sentence there. 

Concerning the comment 3, as the elements mentioned correspond to information extracted from different sources which do not overlap in the detail and elements provided, we believe that keeping the text as a paragraph is more adequate than opting for a table in this early phase of the paper. We revised the corresponding sentences to correct the writing and check for redundancies and we believe the section offers now a more clear reading.

About the information that was provided in the Materials and Methods section, concerning how the literature review was conducted we followed the recommendation of the reviewer and we  eliminated the information about the process of selection and extraction of literature references that were in the lines 261-271 from the previous version of the manuscript. We thank the reviewer for the observation. In the version previously submitted, we had included this information for the sake of clarity and replicability about the process we followed for selecting the papers, and because often this is explicitly required in some publications. We didn't mean to be overwhelming the reader, and we thank you for the observation.

Concerning the observations offered in point 5, about the "Results and Discussion", we performed a through review of the English, and several changes were inserted in the text, leading to a more clean and correct writing. We thank the reviewer and the editor for the opportunity to do so, for the sake of quality of the manuscript. Moreover we inserted some observations about the importance of combining an effective recycling plant, with the deployment of market strategies that allow for the application of the resulting recycled and by-products into ends that are able to provide a financial return for them.

Round 2

Reviewer 3 Report

Reviewrs’ comments have been addressed.